# Recovery of consciousness and cognition after general anesthesia in humans

George A Mashour[1]*, Ben JA Palanca[2], Mathias Basner[3], Duan Li[1], Wei Wang[4], Stefanie Blain-Moraes[1], Nan Lin[4], Kaitlyn Maier[3], Maxwell Muench[2], Vijay Tarnal[1], Giancarlo Vanini[1], E Andrew Ochroch[3], Rosemary Hogg[3], Marlon Schwartz[3], Hannah Maybrier[2], Randall Hardie[3], Ellen Janke[1], Goodarz Golmirzaie[1], Paul Picton[1], Andrew R McKinstry-Wu[3], Michael S Avidan[2]†, Max B Kelz[3]†

[1]Center for Consciousness Science, Department of Anesthesiology, University of Michigan Medical School, Ann Arbor, United States; [2]Department of Anesthesiology, Washington University School of Medicine, St. Louis, United States; [3]Department of Anesthesiology and Critical Care, Perelman School of Medicine, University of Pennsylvania, Philadelphia, United States; [4]Department of Mathematics and Statistics, Washington University, St. Louis, United States

**Abstract** Understanding how the brain recovers from unconsciousness can inform neurobiological theories of consciousness and guide clinical investigation. To address this question, we conducted a multicenter study of 60 healthy humans, half of whom received general anesthesia for 3 hr and half of whom served as awake controls. We administered a battery of neurocognitive tests and recorded electroencephalography to assess cortical dynamics. We hypothesized that recovery of consciousness and cognition is an extended process, with differential recovery of cognitive functions that would commence with return of responsiveness and end with return of executive function, mediated by prefrontal cortex. We found that, just prior to the recovery of consciousness, frontal-parietal dynamics returned to baseline. Consistent with our hypothesis, cognitive reconstitution after anesthesia evolved over time. Contrary to our hypothesis, executive function returned first. Early engagement of prefrontal cortex in recovery of consciousness and cognition is consistent with global neuronal workspace theory.

**\*For correspondence:**
gmashour@umich.edu

†These authors contributed equally to this work

**Competing interests:** The authors declare that no competing interests exist.

## Introduction

The recovery of neurocognitive function after brain network perturbations such as sleep, general anesthesia, or disorders of consciousness is of both scientific and clinical importance. Scientifically, characterizing recovery processes after such perturbations might provide insight into the more general mechanisms by which consciousness and cognition are reconstituted after major network disruptions. The ability to recover cognitive function quickly after sleep, for example, likely confers a natural selection advantage. Moreover, understanding which brain functions are most resilient to perturbation could inform evolutionary neurobiology (*Mashour and Alkire, 2013*; *Kelz and Mashour, 2019*). Clinically, understanding the specific recovery patterns after pathologic states of unconsciousness could inform prognosis or therapeutic strategies. However, it is challenging to characterize differential cognitive recovery after sleep because of the rapidity of the process, whereas it can be impossible in pathologic states because of the unpredictable recovery. General anesthesia, by contrast, represents a controlled and reproducible method by which to perturb consciousness and cognition that is also amenable to systematic observations of the recovery process. Studying recovery of cognition after general anesthesia in humans is also of particular importance because animal studies suggest that general anesthetics have the potential to immediately and persistently impair cognition in the post-anesthetic period (*Culley et al., 2004*; *Valentim et al., 2008*;

**eLife digest** Anesthesia is a state of reversable, controlled unconsciousness. It has enabled countless medical procedures. But it also serves as a tool for scientists to study how the brain regains consciousness after disruptions such as sleep, coma or medical procedures requiring general anesthesia.

It is still unclear how exactly the brain regains consciousness, and less so, why some patients do not recover normally after general anesthesia or fail to recover from brain injury. To find out more, Mashour et al. studied the patterns of reemerging consciousness and cognitive function in 30 healthy adults who underwent general anesthesia for three hours.

While the volunteers were under anesthesia, their brain activity was measured with an EEG; and their sleep-wake activity was measured before and after the experiment. Each participant took part in a series of cognitive tests designed to measure the reaction speed, memory and other functions before receiving anesthesia, right after the return of consciousness, and then every 30 minutes thereafter. Thirty healthy volunteers who did not have anesthesia also completed the scans and tests as a comparison group.

The experiments showed that certain normal EEG patterns resumed just before a person wakes up from anesthesia. The return of thinking abilities was an extended, multistep process, but volunteers recovered their cognitive abilities to nearly the same level as the volunteers within three hours of being deeply anesthetized. Mashour et al. also unexpectedly found that abstract problem-solving resumes early in the process, while other functions such as reaction time and attention took longer to recover. This makes sense from an evolutionary perspective. Sleep leaves individuals vulnerable. Quick evaluation and decision-making skills would be key to respond to a threat upon waking.

The experiments confirm that the front of the brain, which handles thinking and decision-making, was especially active around the time of recovery. This suggests that therapies targeting this part of the brain may help people who experience loss of consciousness after a brain injury or have difficulties waking up after anesthesia. Moreover, disorders of cognition, such as delirium, in the days following surgery may be caused by factors other than the lingering effects of anesthetic drugs on the brain.

*Carr et al., 2011*; *Callaway et al., 2012*; *Zurek et al., 2012*; *Jevtovic-Todorovic et al., 2013*; *Zurek et al., 2014*; *Avidan and Evers, 2016*; *Jiang et al., 2017*), creating a potential public health concern for the hundreds of millions of surgical patients undergoing general anesthesia each year (*Weiser et al., 2015*).

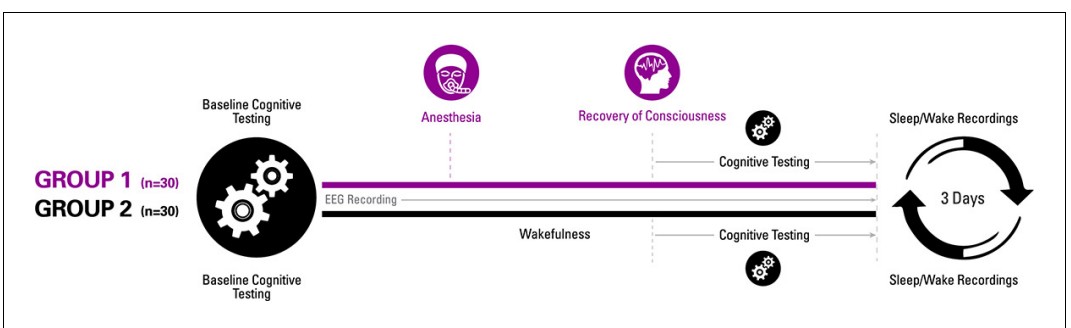

**Figure 1.** Experimental design. Participants were randomized to one of two groups for investigating recovery of consciousness and cognition after general anesthesia. Sleep-wake actigraphy data were acquired in the week leading up to the day of the experiment, which started with baseline cognitive testing followed by either a period of general anesthesia (1.3 age-adjusted minimum alveolar concentration of isoflurane) or wakefulness. Upon recovery of consciousness (or similar time point for controls), recurrent cognitive testing was performed for 3 hr. Actigraphy resumed for 3 days after the experiment.

To improve scientific understanding of recovery of consciousness and cognition after anesthetic-induced unconsciousness, we studied 30 healthy volunteers at three centers who were administered deep general anesthesia using isoflurane for 3 hr, with cognitive testing conducted at pre-anesthetic baseline as well as every 30 min for 3 hr after return of consciousness (*Figure 1*). We hypothesized that post-anesthetic recovery would be an extended process rather than a single point, commencing with return of responsiveness and concluding with return of executive function. We hypothesized that executive function would be the last to recover because there is evidence that neurologic recovery from general anesthesia occurs in a caudal-to-rostral direction (*Långsjö et al., 2012*; *Reshef et al., 2019*), suggesting that anterior structures mediating higher cognition would have the most prolonged recovery. To assess differential return of cognitive functions after the major perturbation of deep anesthesia, we assessed a neurocognitive battery of tests (including the Psychomotor Vigilance Test (PVT), Motor Praxis (MP), Digit Symbol Substitution Test (DSST), fractal 2-Back (NBCK), Visual Object Learning Test (VOLT), and Abstract Matching (AM); *Table 1*) at baseline and at multiple time points after the 3-hr period of anesthetic exposure. Isoflurane anesthesia was chosen because of its heterogeneous molecular targets, which affect multiple neural systems, and because its slower offset compared to other anesthetics would allow us to observe differential recovery of function (*Hemmings et al., 2019*). A halogenated ether was chosen instead of propofol because of the greater diversity of molecular targets, which would be predicted to have a more profound effect on neural dynamics through multiple neurotransmitter receptor and channel systems. Clinical observations as well as clinical research comparing recovery from isoflurane vs. propofol support this interpretation (*Pollard et al., 1994*; *Geng et al., 2017*). The 3-hr duration of anesthesia was chosen based on clinical data related to recovery of surgical patients, the pharmacokinetics of isoflurane, and practical considerations for volunteers participating in day-long experiments.

To control for the learning effects of repeated cognitive testing (*Basner et al., 2018*), we also recruited 30 healthy volunteers who, instead of receiving anesthesia, were engaged in wakeful behavior for 3 hr and then underwent equivalent cognitive testing at time points corresponding to the cohort that underwent general anesthesia (*Figure 1*). All participants received actigraphy watches to monitor sleep-wake activity before and after anesthesia or the control condition, and all participants had electroencephalographic recording throughout the experiment to assess cortical dynamics of relevance to consciousness and cognition, with techniques that have been used to assess information processing in specific brain regions (permutation entropy) as well as more complex spatiotemporal patterns across the cortex (Lempel-Ziv complexity). With the control group serving as a reference, the aims of the study were: (1) to determine whether emergence and cognitive recovery occurred at a point or, as we hypothesized, through a process; (2) assess the sequence of cognitive recovery following emergence from a prolonged state of unconsciousness with serial neurobehavioral assessments to test the hypothesis that higher executive functions reconstitute only after more primary functions; and (3) to measure correlated changes in cortical dynamics that might account for the hypothesized differential recovery of cognitive function.

## Materials and methods
Full methods of the Reconstructing Consciousness and Cognition (ReCCognition) study (clinicaltrials.gov NCT01911195) have been published and are freely available (*Maier et al., 2017*). The published

**Table 1.** Neurocognitive battery, associated cognitive domains, and neuroanatomy.

| Test | Cognitive domains assessed | Brain regions primarily recruited |
| --- | --- | --- |
| Motor Praxis | Sensorimotor speed | Sensorimotor cortex |
| Visual Object Learning | Spatial learning and memory | Medial temporal cortex, hippocampus |
| Fractal 2-Back | Working memory | Dorsolateral prefrontal cortex, cingulate, hippocampus |
| Abstract Matching | Abstraction, concept formation | Prefrontal cortex |
| Digit Symbol Substitution | Complex scanning and visual tracking, working memory | Temporal cortex, prefrontal cortex, motor cortex |
| Psychomotor Vigilance | Vigilant attention | Prefrontal cortex, motor cortex, inferior parietal and some visual cortex |

protocol includes details related to participants, electroencephalography, and neurocognitive testing.

## Ethics

This multicenter study was reviewed and approved by the Institutional Review Board specializing in human subjects research at the University of Michigan, Ann Arbor; University of Pennsylvania; and Washington University in St. Louis. Volunteers were recruited through the use of fliers and were compensated for their participation at levels approved by ethics committees. Participation eligibility required that all subjects provide written informed consent, which was obtained after careful discussion, in accordance with the Declaration of Helsinki.

## Experimental design

The experimental design and data acquisition are summarized in *Figure 1*. This was a within-group study of anesthetized participants with a primary outcome of the pattern of cognitive recovery after general anesthesia; a non-anesthetized cohort was included to control for the learning effects of repeated cognitive testing and circadian factors. Volunteers were randomly assigned to receive general anesthesia with isoflurane or to engage in waking activity on the study day and serve as experimental controls. Baseline cognitive assessment was performed after an initial screening.

For anesthesia sessions, participants were closely monitored by two attending anesthesiologists during the study day. Attending anesthesiologists elicited a standard clinical preoperative history and physical examination, independently verified that volunteers met inclusion and fasting criteria, and safely conducted the anesthetic. Each subject underwent intravenous catheter placement. An appropriately fitted EEG head cap (Electrical Geodesics, Inc Eugene OR) was affixed to the scalp. Electrical impedances on each channel were kept under 50kOhms/channel whenever possible. Standard anesthesia monitors (electrocardiogram, non-invasive blood pressure cuff, a pulse oximeter) were applied, and capnography was measured. Subjects completed a second baseline round of neurocognitive testing with ongoing EEG recordings. Upon completing the neurocognitive battery, subjects were pre-oxygenated by face mask prior to induction of general anesthesia with a stepwise increasing infusion rate of propofol: 100 mcg/kg/min x 5 min, 200 mcg/kg/min x 5 min, and then 300 mcg/kg/min x 5 min. During this time, an audio loop issued commands every 30 s asking subjects to squeeze their left or right hand (in random order) twice. Loss of consciousness (LOC) was defined as the first time that a subject failed to respond to two sets of consecutive commands. After 15 min of propofol administration, subjects began inhaling isoflurane at 1.3 age-adjusted MAC (minimum alveolar concentration) (*Nickalls and Mapleson, 2003*). Thereafter, a laryngeal mask was inserted orally, a nasopharyngeal temperature probe was placed, and the propofol infusion was discontinued. Anesthetized subjects continued to inhale 1.3 age-adjusted MAC isoflurane anesthesia for 3 hr. Burst suppression, a sign of deep anesthesia (*Hemmings et al., 2019*), was found to be associated with this concentration of isoflurane in this cohort (*Shortal et al., 2019*). Blood pressure was targeted to remain within 20% of baseline pre-induction values using a phenylephrine infusion or intermittent boluses of ephedrine, as necessary. Pressure support ventilation was initiated with pressures titrated to maintain tidal volumes in the 5–8 ml/kg range while end tidal carbon dioxide levels were targeted to 35–45 torr. Surface warming blankets were utilized to maintain body temperature in the normal range. Subjects received 4 mg intravenous ondansetron 30 min prior to discontinuation of isoflurane for antiemetic prophylaxis.

Isoflurane was discontinued at the end of the 3-hr anesthetic period. Verbal command loops were reissued every 30 s immediately upon cessation of isoflurane. The laryngeal mask was removed when deemed medically safe by the attending anesthesiologists. Recovery of consciousness (ROC) was defined as the earliest instance in which subjects correctly responded to two consecutive sets of audio loop commands. At this point, defined as time = 0 min, subjects restarted neurocognitive testing with a brief pause between consecutive rounds. Neurocognitive testing was repeated at t = 30, 60, 90, 120, 150, and 180 min following emergence. Each battery of neurocognitive testing lasted approximately 15–25 min and was preceded by 5 min of eyes closed, resting state EEG data acquisition. Brief restroom or nutrition breaks were permitted between testing rounds, as necessary.

Subjects were discharged according to standard post-anesthesia care unit discharge criteria after completing their final battery of neurocognitive testing. A study site coordinator contacted each subject within 24 hr of the study day to document any adverse events.

A second group of healthy individuals (n = 30) was recruited to participate in the same study design. These individuals also fasted overnight, but did not have intravenous lines inserted. Rather than being anesthetized, these volunteers remained awake (by reading or watching television on a personal electronic device) and continued fasting for 3.5 hr in order to control both for potential learning effects of repeated testing and also for circadian variability in testing performance (*McLeod et al., 1982*; *Gur et al., 2001*; *Van Dongen et al., 2003*; *Van Dongen and Dinges, 2005*; *Jasper et al., 2009*; *Tucker et al., 2010*). We chose not to sedate these participants as a control for the anesthetized state because doing so would have obscured the predicted learning effect accompanying repeated neurobehavioral testing and would thus confound the normal performance standard that was required. Volunteers randomized to the restful group were instructed to avoid napping and were regularly monitored by a dedicated research assistant.

## Participants

A total of 60, healthy American Society of Anesthesiologists physical status classification I or II volunteers were enrolled. The choice of study subject numbers was informed by several factors, including previous studies (*Eger et al., 1997*; *Eger et al., 1998*), biological plausibility (our best estimates regarding effect size and standard deviation), and safety considerations (exposing the minimum number of humans to general anesthesia in order to answer the questions of interest). The main factor that was considered in estimating our required sample size was the time difference in return of cognitive functions within the subjects receiving general anesthesia. Sample size calculation was modeled with various assumptions regarding the difference in recovery times between the first and last cognitive domains to return, and the standard deviations of these parameters. A range between 30 min and 90 min was considered for differences in recovery times between cognitive domains (possible effect sizes). A range between 20 min and 40 min was considered for standard deviations of these parameters. Assuming relatively conservative estimates (difference in recovery times = 30 min and standard deviation = 40 min), 30 subjects would provide >80% power with a two-sided alpha <0.05, using an unpaired t test. With relatively liberal assumptions (difference in recovery times = 90 min and standard deviation = 20 min), 30 subjects would provide >99% power with a two-sided alpha <0.001, using an unpaired t test.

Each study site (University of Michigan, University of Pennsylvania, Washington University in St. Louis) recruited 20 volunteers who met the inclusion criteria. Prospective volunteers were screened using a phone questionnaire administered by a study coordinator. Eligible subjects that consented to participate in this study underwent a baseline familiarization round of neurocognitive testing and were given rest-activity monitoring devices (actiwatch) 1 week prior to the study day.

## EEG acquisition and analysis

To assess the neural correlates of the anesthetized state and recovery, participants enrolled at the University of Pennsylvania and University of Washington in St. Louis (n = 40) were fitted with 32 EEG scalp electrodes. Subjects at the University of Michigan (n = 20) were fitted with 64 or 128 EEG scalp electrodes. EEG recordings began prior to the baseline neurocognitive testing on the study day and were continued with minimal interruption until the completion of the final neurocognitive test.

The raw EEG signals were exported into MATLAB (version 2015a; MathWorks, Inc, Natick, MA), down-sampled to 250 Hz (resample.m function in Matlab signal processing toolbox), and re-referenced to the linked-mastoid reference. Electrodes on the lowest parts of the face and head were removed, leaving 21 channels on the scalp (common to EEG montage for all participants) for the analysis. Data segments with obvious noise or non-physiological artifacts were identified and removed by visual inspection of the waveform and spectrogram of the EEG signals. Prior to the analysis, the EEG signals were bandpass filtered at 0.5–30 Hz via butterworth filter of order 4 (butter.m and filtfilt.m in MATLAB signal processing toolbox) to remove the possible baseline drift and muscle artifacts.

Ten 2 min epochs were selected during the seven resting-state eyes-closed sessions, and also during the exposure to anesthesia: (1) LOC - the first 2 min after loss of responsiveness; (2)

Maintenance – the last 2 min before the discontinuation of isoflurane; and (3) Pre-ROC – the 2 min immediately preceding recovery of responsiveness. Burst-suppression patterns were present in the Maintenance epoch for six participants; to prevent the confounding effect of the suppression pattern on the EEG measures, we instead extracted 2 min continuous, non-suppression epochs in the last 10 min before the discontinuation of isoflurane (n = 3 participants), or 7 min immediately after the discontinuation of isoflurane (n = 3 participants), which showed similar spectral properties when compared to the other participants. Detailed information on the EEG sample size is listed in *Supplementary file 1A*. For completeness, seven 2 min epochs were selected during the seven resting-state eyes-closed sessions in the non-anesthetized group.

Preliminary analysis with high-density EEG suggested that phase-based connectivity measures and graph-theoretical variables were not robust enough to capture the neurophysiologic dynamics of general anesthesia and recovery with appropriate temporal resolution (*Blain-Moraes et al., 2017*). We therefore chose two measures of cortical dynamics that have been suggested to differentiate levels of consciousness and, unlike spectral analysis, also serve as a surrogate for the repertoire of brain states, which we would expect to increase with recovery of consciousness and cognitive function.

## Permutation entropy

We used permutation entropy (PE) to measure the local dynamical changes of EEG in frontal and posterior channels. PE quantifies the regularity structure of a time series, based on a comparison of the order of neighboring signal values, which is conceptually simple, computationally efficient, and artifact resistant (*Bandt and Pompe, 2002*), and has been successfully applied to the separation of wakefulness from unconsciousness (*Jordan et al., 2013*; *Li et al., 2008*; *Olofsen et al., 2008*; *Ranft et al., 2016*). The calculation of PE requires two parameters: embedding dimension ($d_E$) and time delay (τ). In line with previous studies, we used $d_E$=5 and τ=4 in order to provide a sufficient deployment of the trajectories within the state space of the EEG beta activity during wakefulness and anesthesia (*Jordan et al., 2013*; *Ranft et al., 2016*). Supplementary analysis was performed to test the sensitivity of PE using alternative strategies of parameter selection.

In the implementation, each 2 min epoch was divided into non-overlapping 10 s windows, the PE was calculated for each window, and the PE values were averaged across all the windows for each studied epoch and channel. The topographic maps of group-level PE value for each studied epoch was constructed using the topoplot function in the EEGLAB toolbox (*Delorme and Makeig, 2004*). For statistical comparisons, the averaged PE values were calculated over the frontal (Fp1, Fp2, Fpz, F3, F4, and Fz) and posterior (P3, P4, Pz, O1, O2, and Oz) channels at each studied epoch for each participant.

## Lempel-Ziv complexity

Lempel-Ziv Complexity (LZC) was computed as a surrogate of complexity to reflect the spatiotemporal repertoire across scalp potentials. LZC is a method of symbolic sequence analysis that measures the complexity of finite length sequences (*Lempel and Ziv, 1976*), which has been shown to be a valuable tool to investigate brain states related to consciousness and cognition (*Casali et al., 2013*; *Abásolo et al., 2015*; *Schartner et al., 2015*; *Hudetz et al., 2016*; *Schartner et al., 2017*). The calculation of LZC requires a binarization of the multichannel EEG data. In this study, we used the implementation as described in *Schartner et al., 2015*; *Schartner et al., 2017*, and calculated the instantaneous amplitude from the Hilbert transformed EEG signal for each channel, which was binarized using its mean value as the threshold for the current channel (supplementary analysis was performed to test the effect of threshold selection). The data segment was then converted into a binary matrix, in which rows represent channels and columns represent time points. LZC was computed by searching the spatiotemporal matrix time point by time point and counting the number of different spatial patterns across different time points. Thus, the resultant measure captures the complexity or diversity in both temporal and spatial domains.

For implementation, the average signal was first subtracted from all channels in order to remove the effect of common reference, and then the multichannel EEG epochs were divided into non-overlapping 4 s windows to compute the LZC, with the resultant LZC values being averaged across all the windows for each studied epoch. In line with previous studies, we normalized the original LZC by

the mean of the LZC values from N = 50 surrogate data sets generated by randomly shuffling each row of the binary matrix, which is maximal for a binary sequence of fixed length (*Schartner et al., 2015*; *Schartner et al., 2017*) (supplementary analysis was performed to test alternative methods in the generation of surrogate data).

## Statistical analysis of EEG measures

Statistical analyses were conducted in consultation with the Center for Statistical Consultation and Research at the University of Michigan. All EEG-derived PE and LZC values were exported to IBM SPSS Statistics version 24.0 for Windows (IBM Corp. Armonk, NY). Statistical comparisons were performed using linear mixed models (LMM), to test (1) the difference between the ten studied epochs for both PE and LZC measures and (2) the difference between PE values derived from frontal and posterior channels. In contrast to traditional repeated-measures ANOVA analysis, LMM analysis offers more flexibility in dealing with missing values (see *Supplementary file 1B*) and accounting for the within-participant variability by including a random intercept associated with each participant. The non-anesthetized group was included primarily to control for learning effects in repeated cognitive testing and thus, for EEG analysis, the statistical analysis was focused on anesthetized group. For the model of LZC values, the fixed effect is the studied epoch. For the model of PE values, the fixed effects include the studied epoch, region, and the interaction between them. We fitted the models with random intercept specific for each participant and used the default variance components as the covariance structure. We modeled the studied epoch as repeated effects and assumed each studied epoch was associated with different residual variance by using the diagonal structure as the covariance structure of the residuals. We employed restricted maximum likelihood estimation. The models described above were chosen by taking into account the information criteria and likelihood ratio test results in the comparisons, with alternative models including additional random effects and repeated effects, as well as different covariance structures (*Supplementary file 1B*). For all post hoc pairwise comparisons, the Bonferroni corrected p-value along with the estimate and 95% confidence interval (CI) of the difference were reported. A two-sided p<0.05 was considered statistically significant.

## Associations between EEG measures and neurocognitive performance

To explore whether the EEG-based measures of cortical dynamics are predictive of the performance of the cognitive functions in the post-anesthetic period, we examined the associations between EEG measures during pre-anesthetic baseline (EC1), Maintenance and pre-ROC and the impairment of cognitive performance at emergence (just after recovery of consciousness). Spearman's rank correlation was used to assess the relationships between each of the EEG measures (frontal PE, posterior PE and LZC) and the impairment of performance, in terms of both accuracy and response time, for each of the six tasks.

## Neurocognitive testing

Neurocognitive tests were selected from the cognition test battery (*Basner et al., 2015*) to reflect a broad range of cognitive domains, ranging from basic abilities such as sensory-motor speed to complex executive functions such as abstraction. The order of the six tests was randomized but balanced across subjects. Individual subjects took the tests in the same order (except during familiarization, which occurred at least 1 week prior to the study day). In each test session, subjects repeated the first test after completion of sixth test. Therefore, the temporal resolution for one test in five control subjects and five experimental subjects was doubled. Each testing session required 15–25 min to complete, with a new session starting every 30 min after emergence. There was thus a total of six testing sessions in the 3-hr time period after emergence. The following six tests, adopted from *Basner et al., 2015*, were chosen for this study. See *Table 1* for a summary of the tests, associated cognitive function, and associated neuroanatomical substrates.

The *Motor Praxis Task (MP)* measures sensorimotor speed and validates that volunteers have sufficient command of the computer interface. Participants were instructed to click on squares that appear randomly on the screen, with each successive square smaller and thus more difficult to track. The test depends upon function of visual and sensorimotor cortices (*Gur et al., 2001*; *Gur et al., 2010*; *Neves et al., 2014*).

The *Psychomotor Vigilance Test (PVT)* measures a volunteer's reaction times (RT) to visual stimuli that are presented at random inter-stimulus intervals over 3 min (*Basner et al., 2011*). Subjects monitor a box on the computer screen, and press the space bar once a millisecond counter appears and begins timing response latency. In the well-rested state, or whenever sustained attention performance is optimal, right frontoparietal cortical regions are active during this task. Conversely, with sleep deprivation and other suboptimal performance, studies demonstrate increased activation of default-mode networks during this task, which is considered to be a compensatory mechanism (*Drummond et al., 2005*).

The *Digit-Symbol Substitution Task (DSST)* adapts the Wechsler Adult Intelligence Scale (WAIS-III) for a computerized presentation. The DSST required participants to refer to a continuously displayed legend that matches each numeric digit to a specific symbol. Upon presentation of one of the nine symbols, subjects must select the corresponding number as rapidly as possible. The DSST primarily recruits the temporal cortex, prefrontal cortex, and motor cortex. Activation of frontoparietal cortices during DSST performance has been interpreted as reflecting both on-board processing in working memory and low-level visual search (*Usui et al., 2009*).

The *Fractal 2-Back (F2B)* is an extremely challenging nonverbal variant of the Letter 2-Back test. N-back tests probe working memory. The F2B consisted of the sequential presentation of a set of fractals, each potentially repeated multiple times. Participants were instructed to respond when the current stimulus matched the stimulus previously displayed two images prior. The F2B is a well-validated task that robustly activates dorsolateral prefrontal cortex, cingulate, and hippocampus (*Ragland et al., 2002*).

The *Visual Object Learning Test (VOLT)* measures the volunteer's memory for complex figures (*Glahn et al., 1997*). Participants memorize 10 sequentially displayed three-dimensional figures. Subsequently, they were instructed to select the familiar objects that they memorized from a larger set of 20 sequentially presented objects that included the 10 memorized and 10 similar but novel objects. Visual object learning tasks have been shown to depend upon frontal and bilateral anterior medial temporal cortices as well as the hippocampus (*Jackson and Schacter, 2004*).

The *Abstract Matching (AM)* test (*Glahn et al., 2000*) is a validated test of executive function. Subjects are presented with two pairs of objects at the bottom left and right of the screen whose perceptual dimensions (e.g. color and shape) vary. Subjects were presented with a target object in the upper middle of the screen and had to classify the target using its perceptual dimensions to one of the two pairs, based on a set of implicit, abstract rules. The abstract matching paradigm evaluates abstraction and cognitive flexibility and depends upon the prefrontal cortex (*Berman et al., 1995*).

## Statistical analysis of cognitive data

Apart from the Bayesian analyses, statistical analyses were implemented by the SAS software version 9.4. In view of the learning effect with repeated cognitive testing, it is difficult to pinpoint when any cognitive domain returns to baseline. We could therefore not simply compare recovery times between individual cognitive tests, as we had planned. Instead, in order to test whether there was a difference in recovery times between cognitive domains, we opted for a Bayesian regression approach using all the data from the anesthetized subjects as well as the non-anesthetized controls. For this analysis, we used the brms package in R (R Foundation for Statistical Computing, Vienna, Austria), which uses Stan for full Bayesian inference. We simulated M samples from the (posterior) conditional distribution of the model parameters given the data, and for each set of simulated model parameters, we calculated the corresponding recovery time, thus we obtained M samples of recovery times. Then, based on these simulated recovery times, we further calculated the corresponding differences in recovery times between pairs of cognitive tests (or domains), and evaluated the posterior probability of one cognitive test recovering more than 30 min before another test [P(diff >30 min|data)] by checking the sample proportion in the posterior sample. The recovery time involves both performance accuracy and performance speed. Both processes are defined by the time for anesthetized subjects to have the test score back to their respective accuracy/speed baseline scores. Regarding priors, we chose normal priors with a large variance, which is a routine choice for non-informative flat priors.

Multiple statistical comparisons were separately conducted on the standardized accuracy and speed indices, which were two metrics to evaluate each task performance. We used nonlinear mixed-effects models (NLMM) based on a damped exponential in time to fit the data of each task at

all time-points, that is $y_t = y_{baseline} + \alpha + \beta * e^{\gamma t}$, where $y_t$ is the task performance response at time $t$, $y_{baseline}$ is the pre-treatment baseline response, $\alpha$ is the random intercept, $\beta$ is the random slope, and $\gamma$ is the coefficient of the fixed-effect time $t$. Based on our model, the recovery time was calculated by $T = \log\left(-\frac{\alpha}{\beta}\right)/\gamma$. The random intercept $\alpha$ and the random slope $\beta$ are independent, and distributed from normal distributions. Some NLMMs were degenerated to models with a fixed intercept or a fixed slope according to goodness of fit. The appeal of NLMM analysis is its flexibility in modeling the nonlinear trend of cognitive data with repeated measures over time and the ability to adjust the pre-treatment baseline performance. In order to test against the alternative hypothesis that there was a significant difference at the end of the 3-hr period between the anesthetized group and the control group, we performed model-based multiple testing on the least squares means of predicted response difference at 3 hr. For accuracy and speed of each task, the point estimates of the difference as well as the Bonferroni corrected confidence intervals (CI) of the difference with an overall significance level 0.05 were reported. The p-values of testing difference between groups should be compared to the corrected level 0.05/6=0.0083. For both accuracy and speed estimates of recovery time (measured in hours) in each testing domain, we based our estimates on 10,000 Markov Chain Monte Carlo samples. For these simulations, we present posterior probabilities and corresponding 90% credible intervals for differences between cognitive domains in recovery times.

## Actigraphy

In order to assess and control for differences in baseline sleep-wake rhythms and to potentially evaluate the effect of isoflurane anesthesia on subsequent rest activity behavior, participants were trained and instructed to wear a wrist GT3X + device (ActiGraph) on their nondominant wrist beginning at the conclusion of their baseline visit, 1 week before prior to and 1 week following their assigned study day. Actigraphy data were downloaded to a computer and GT3X + devices recharged on the study day and again at completion 1 week following the study day. Raw activity counts for each subject were binned into 1 min epochs and analyzed for bouts of inactivity using the Cole-Kripke scoring algorithm (*Cole et al., 1992*), included in the ActiLife 6.7.2 software. For each subject, minutes of inactivity each hour were calculated. ActiLife's wear time validation was employed using default settings to confirm that subjects used the GT3X + monitor as instructed. Hours in which wear time validation revealed that the watch was not worn were excluded from the analysis.

## Statistical analysis of actigraphy data

Actigraphy data were imported into Prism 5.0d (GraphPad) and analyzed with a two-way ANOVA with Time in hours relative to the midnight before testing and Treatment Group (Isoflurane Exposed or Awake Controls) as the two factors. Effects of Time, Treatment Group, and the interaction between these two factors were considered to be significant for p values < 0.05. Due to asynchrony in times during which the GT3X + device was not worn across individuals, it was not possible to conduct a repeated measures two-way ANOVA. To obtain a graphical best fit of actigraphy data, we performed a standard Cosinar analysis.

# Results

## Participants

The study received ethics committee approval at all three sites independently; written informed consent was obtained after careful discussion with each participant. The average age of all study participants was 27 (±4.5) years, with 50% females. There were no adverse events during the course of the study or at 1-week follow up at the completion of the study.

## Recovery of cognitive functions after general anesthesia

We administered six distinct cognitive tests at baseline and twice per hour for 3 hr after exposure to general anesthesia or a comparable period without anesthetic exposure in the control participants. The order of the MP, PVT, DSST, NBCK, VOLT, and AM tests was randomized between subjects but consistent within subjects. In anesthetized volunteers (with correction for learning based on results

from tests taken at corresponding times by the non-anesthetized controls), the accuracy and speed of all six cognitive tests were significantly impaired compared with the pre-anesthetic baseline assessments (all twelve statistical tests yielded p<0.008, with adjustments for multiple comparisons). Thus, the first question answered is that all tests were impaired at initial recovery.

The next question to be answered was whether rates of recovery differed among cognitive domains and whether the time to recovery exceeded 30 min when comparing among cognitive tests. Based on likelihood ratio tests, there were statistically significant differences in the rates of recovery of the six cognitive domains, both with regard to their accuracy and speed (p<0.05). Results from Bayesian analyses yielded posterior probability estimates that differences in recovery times between the various cognitive domains for accuracy and speed exceeded 30 min. Hence, recovery of cognition is a process that appears to evolve over time rather than one that occurs simultaneously.

For accuracy (*Figure 2*), the probability was high that (i) recovery of AM occurred more than 30 min before NBCK (95%), DSST (77%), VOLT (72%), and MP (65%); (ii) recovery of PVT occurred more than 30 min before NBCK (99.9%), DSST (81%), MP (76%), and VOLT (72%); (iii) recovery of VOLT occurred more than 30 min before NBCK (95%), DSST (76%), and MP (68%); and that (iv) recovery of MP occurred more than 30 min before NBCK (57%).

The results from the 10,000 Markov Chain Monte Carlo samples yield the following additional insights with respect to accuracy of recovery. The accuracy in PVT is strongly impaired as most samples cannot recover. For the other five tests, the posterior probability that: (a) recovery of MP occurred more than 30 min before VOLT, NBCK and DSST are 93%, 97%, and 99%, respectively; the 90% corresponding credible intervals of the difference in recovery time are (−2.82,–0.27), (−3.6,–0.77), and (−0.15, 0.78); (b) recovery of VOLT occurred more than 30 min before NBCK and DSST are 55% and 50%, respectively; the 90% corresponding credible intervals of the difference in recovery time are (−2.38, 1.32) and (−1.69, 0.74); (c) recovery of AM occurred more than 30 min before VOLT, NBCK, and DSST are 98%, 99%, and 100%, respectively; the 90% corresponding credible intervals of the difference in recovery time are (−2.92,–0.47), (−3.66,–0.91), and (−2.47,–1.15); (d)

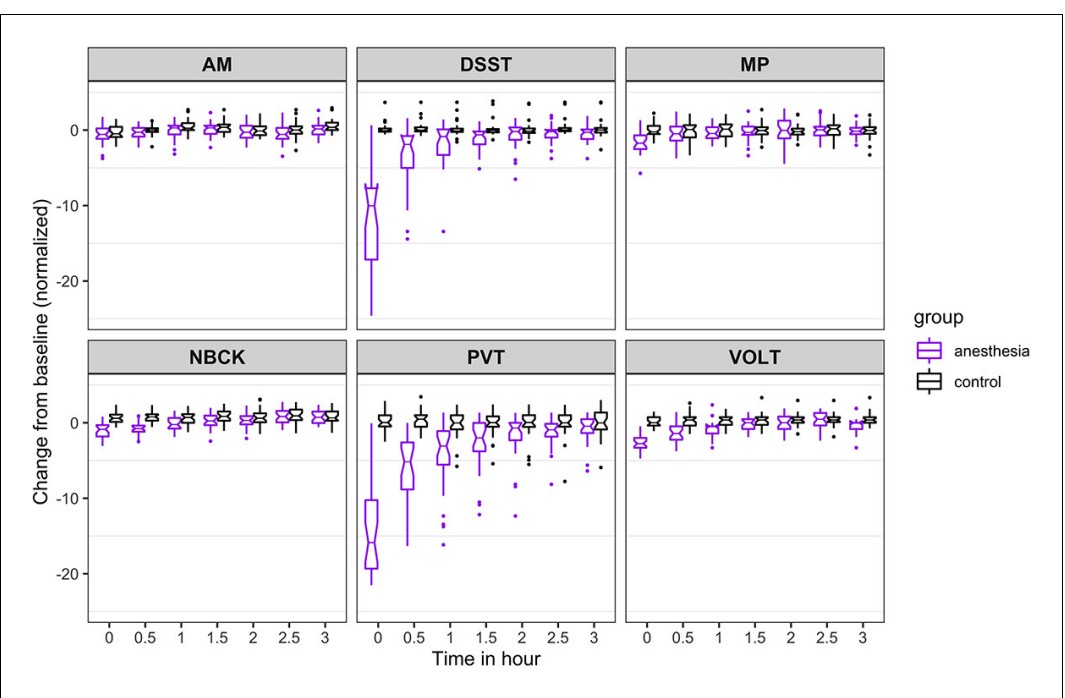

**Figure 2.** Time course for recovery of (normalized) accuracy in cognitive task performance after general anesthesia (time 0 is just after recovery of consciousness in the group that was anesthetized). AM, Abstract Matching; DSST, Digit Symbol Substitution Test; MP, Motor Praxis; NBCK, Fractal 2-Back; PVT, Psychomotor Vigilance Test; VOLT, Visual Object Learning Test. The six cognitive tests are all represented.

The online version of this article includes the following source data for figure 2:

**Source data 1.** Source data supporting *Figure 2*.

recovery of DSST occurred more than 30 min before NBCK is 42%; the 90% corresponding credible interval of the difference in recovery time is (−1.55, 1.09).

For speed (*Figure 3*), the probability was high that (i) recovery of AM occurred more than 30 min before PVT (68%), VOLT (64%), and DSST (52%); (ii) recovery of NBCK occurred more than 30 min before PVT (74%), VOLT (66%), and DSST (63%); (iii) recovery of MP occurred more than 30 min before PVT (81%); and that (iv) recovery of DSST occurred more than 30 min before PVT (53%). We had further hypothesized that, if there were differential rates of recovery in cognitive domains, higher order executive function (as tested by AM) would be most impaired and consequently would be the last to recover. Contrary to this expectation, AM was one of the least impaired of the tests at ROC. For both accuracy and speed measures, performance on AM quickly approached its pre-anesthetic level.

The results from the 10,000 Markov Chain Monte Carlo samples yield the following additional insights with respect to speed of recovery. The recovery of NBCK (as measured by speed to task completion) is strongly impaired. Most samples did not recover over the experimental time course, a fact mirrored in the simulations. For the other five tests, the posterior probability that: (a) recovery of MP speed occurred more than 30 min before AM, PVT, and DSST are 54%, 91%, and 83%, respectively; the 90% corresponding credible intervals of the difference in recovery time are (−1.31, 0.24), (−3.95,–0.02), and (−1.64,–0.24); (b) recovery of VOLT speed occurred more than 30 min before MP, AM, PVT, and DSST are 82%, 98%, 100%, and 100%, respectively; the 90% corresponding credible intervals of the difference in recovery time are (−1.97,–0.12), (−2.35,–0.79), (−4.84,–1.17), and (−2.66,–1.33); (c) recovery of AM speed occurred more than 30 min before PVT and DSST are 79% and 40%, respectively; the 90% corresponding credible intervals of the difference in recovery time are (−3.49, 0.10) and (−0.92, 0.10); (d) recovery of DSST occurred more than 0.5 hr before PVT is 64%; the 90% corresponding credible interval of the difference in recovery time is (−3.02, 0.47).

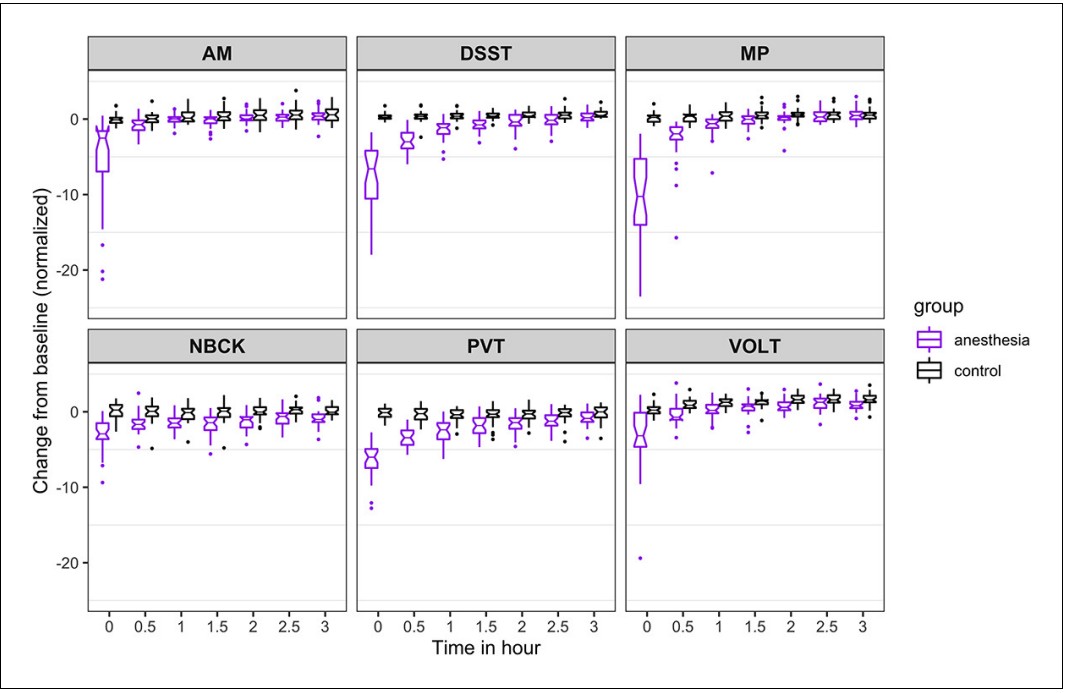

**Figure 3.** Time course for recovery of (normalized) speed of cognitive task performance after general anesthesia (time 0 is just after recovery of consciousness in the group that was anesthetized). AM, Abstract Matching; DSST, Digit Symbol Substitution Test; MP, Motor Praxis; NBCK, Fractal 2-Back; PVT, Psychomotor Vigilance Test; VOLT, Visual Object Learning Test. The six cognitive tests are all represented.

The online version of this article includes the following source data for figure 3:

**Source data 1.** Source data supporting *Figure 3*.

As expected, for all tests, the maximal degree of impairment was upon ROC. Information on the trajectory of recovery for an individual cognitive test is depicted in *Figures 2* and *3*, and was assessed using a non-linear mixed effects model. In the 3 hr follow-up period, accuracy for those in the anesthesia group increased gradually for some tests (DSST, PVT) and more rapidly for others (AM, NBCK, MP, VOLT). At 3 hr after ROC in the anesthetized group, accuracies in five out of six tests were not statistically significantly different from the control group. There remained a significant (p<0.001), albeit small, difference in accuracy performance between the anesthetized and the control group on the VOLT at 3 hr. Overall, within 3 hr of return of consciousness, the anesthetized group returned to an accuracy level that was not substantially different from that of participants who were not anesthetized. The comparison with the non-anesthetized control group is highly informative, because assessing performance on cognitive tests over time is confounded by learning. Had the anesthetized group simply returned to, or even exceeded, their own baseline performance after 3 hr of testing, this would not have provided sufficient evidence to conclude that their cognition had truly returned to baseline. The inclusion of an awake control group strengthens the conclusion that the anesthetized group did not experience a decrement in their performance that might have been masked by learning, which is known to occur with repeated testing.

To account for a trade-off in accuracy versus speed, we also evaluated speed of task performance (*Figure 3*). Speed was also most strongly impaired at ROC, with a drop of more than 5 SD for all tests but the NBCK. At 3 hr after ROC in the anesthetized group, speed in two (MP and PVT) out of six tests were not statistically significantly different from the control group. There remained significant (p<0.001), albeit small, differences between the anesthetized and the control group in speed performance on four (VOLT, NBCK, AM, DSST) tests at 3 hr. The results of the nonlinear mixed-effects models for cognitive performance in anesthetized and non-anesthetized cohorts are summarized in *Table 2*.

## Cortical dynamics before, during, and after general anesthesia

We assessed cortical dynamics before, during, and after anesthetic exposure using local measures of permutation entropy (PE) and global measures of Lempel-Ziv complexity (LZC). Using a linear mixed model, the PE demonstrated significant differences associated with behavioral states ($F_{9, 86}$ = 42.423, p<0.001), brain regions ($F_{1, 257}$ = 4.275, p=0.040), and the interaction between them ($F_{9, 85}$ = 2.750, p=0.007). As compared to the baseline condition of eyes-closed resting state, frontal PE decreased at propofol-induced loss of consciousness (LOC), further decreased during maintenance of the anesthetized state with isoflurane anesthesia (p<0.001,−0.160 [-0.185 to −0.135], maintenance vs. EC1), and returned to or even exceeded the baseline level just before the recovery of consciousness (ROC) (p=0.002, 0.036 [0.018 to 0.055], pre-ROC vs. EC1) (*Figure 4A and B*). Posterior PE did not show significant changes at LOC but was decreased during the maintenance phase (p<0.001,−0.110 (-0.135 to −0.085), maintenance vs. EC1), and then returned to baseline level just before ROC

**Table 2.** Results of nonlinear mixed-effects models comparing cognitive trajectories at 3 hr post emergence between anesthetized and non-anesthetized cohorts.

AM, Abstract Matching; DSST, Digit Symbol Substitution Test; MP, Motor Praxis; NBCK, Fractal 2-Back; PVT, Psychomotor Vigilance Test; VOLT, Visual Object Learning Test. For speed and accuracy of each task, we report the Bonferroni corrected confidence intervals (CI) of the difference with an overall significance level 0.05. The p-values of testing difference between groups should be compared to the corrected level 0.05/6 = 0.0083.

| | Speed | | | Accuracy | | |
|---|---|---|---|---|---|---|
| | **Estimate** | **p-value** | **CI** | **Estimate** | **p-value** | **CI** |
| MP | 0.3562 | 0.1101 | (−0.2437, 0.9561) | 0.0366 | 0.8721 | (−0.5824, 0.6556) |
| VOLT | 0.7666 | 0.0720 | (−0.3757, 1.9089) | 0.3915 | 0.0847 | (−0.2184, 1.0015) |
| NBCK | 1.2803 | <0.0001 | (0.6642, 1.8964) | 0.0089 | 0.9664 | (−0.5679, 0.5857) |
| AM | 0.6336 | 0.0023 | (0.09021, 1.1770) | 0.3692 | 0.0709 | (−0.1791, 0.9176) |
| PVT | 0.7986 | 0.0026 | (0.1052, 1.4921) | 1.0916 | 0.0557 | (−0.4359, 2.6191) |
| DSST | 1.0184 | <0.0001 | (0.4420, 1.5949) | 0.5454 | 0.0931 | (−0.3279, 1.4188) |

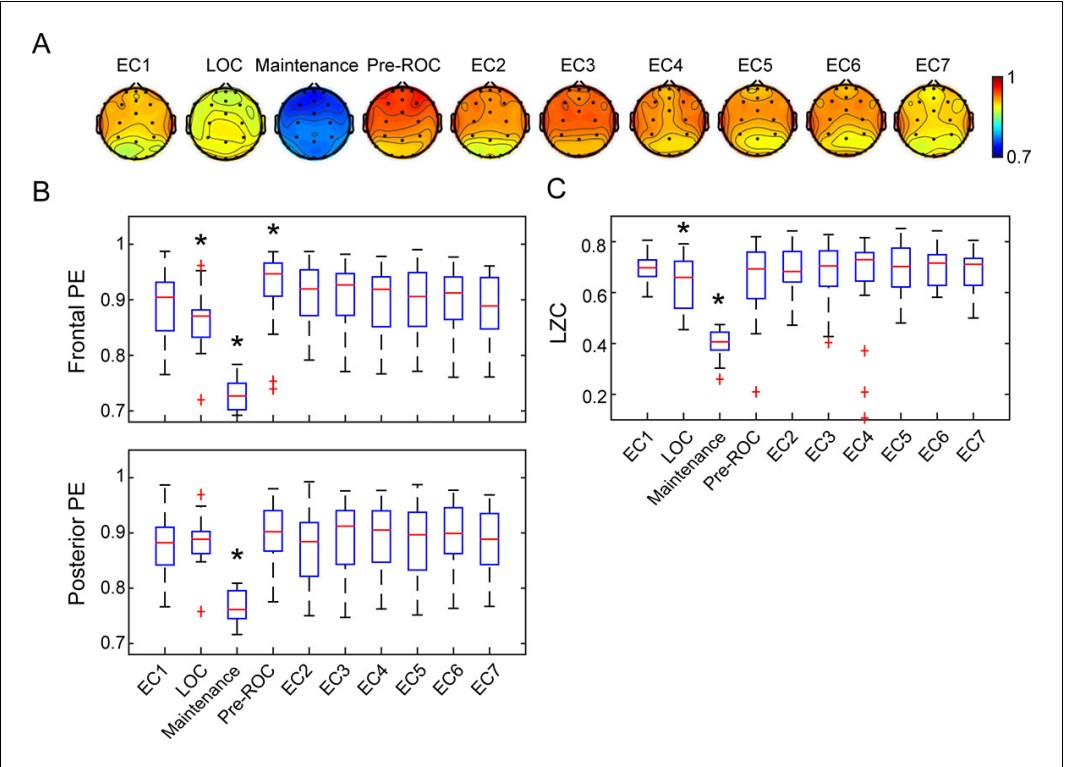

**Figure 4.** Cortical dynamics before, during, and after general anesthesia. (**A**) Scalp topographic maps of the group-level permutation entropy (PE; median average across N = 30 participants) at the ten studied epochs. (**B**) The box plots of average PE values in frontal (Fp1, Fp2, Fpz, F3, F4, and Fz) and posterior channels (P3, P4, Pz, O1, O2, and Oz) for the studied epochs. On each box, the central mark is the median, the edges are the 25th and 75th percentiles, the whiskers extend to the most extreme data points determined by the MATLAB algorithm to be non-outliers, and the points deemed by the algorithm to be outliers are plotted individually (red cross). (**C**) The box plots of LZC values for the studied epochs. EC = eyes closed resting state (EC1 is baseline consciousness, EC2-7 are post-emergence just prior to cognitive testing), LOC = loss of consciousness, Pre-ROC = 2 min epoch just before recovery of consciousness. *indicates significant difference relative to EC1, using linear mixed model analysis (Bonferroni-corrected p<0.05).

The online version of this article includes the following source data and figure supplement(s) for figure 4:

**Source data 1.** Source data for *Figure 4A, B*.

**Source data 2.** Source data for *Figure 4C*.

**Figure supplement 1.** Confirmatory results of permutation entropy (PE) with different settings of embedding dimension ($d_E$) and time delay ($\tau$).

**Figure supplement 2.** Confirmatory results of Lempel-Ziv Complexity (LZC).

**Figure supplement 3.** Cortical dynamics as assessed by permutation (PE) and Lempel-Ziv complexity (LZC) for the non-anesthetized control group.

**Figure supplement 4.** Associations between EEG measures during pre-anesthetic baseline (EC1) with the impairment of cognitive functions at emergence (just after recovery of consciousness).

**Figure supplement 5.** Associations between EEG measures during maintenance with the impairment of cognitive functions at emergence.

**Figure supplement 6.** Associations between EEG measures during pre-ROC with the impairment of cognitive functions at emergence.

(*Figure 4A and B*). The topographic maps of PE exhibited region-specific patterns, in which frontal channels demonstrated significantly higher PE values as compared to posterior channels at the eyes-closed resting state directly after emergence (EC2, p=0.002, 0.032 [0.016 to 0.048], frontal vs. posterior) (*Figure 4A*).

The LZC demonstrated significant state-dependent differences ($F_{9,\ 50}$ = 59.364, p<0.001). As compared to baseline consciousness, LZC declined at LOC and decreased further during the maintenance phase (p<0.001,–0.258 [-0.285 to −0.231], maintenance vs. EC1), returning to baseline level

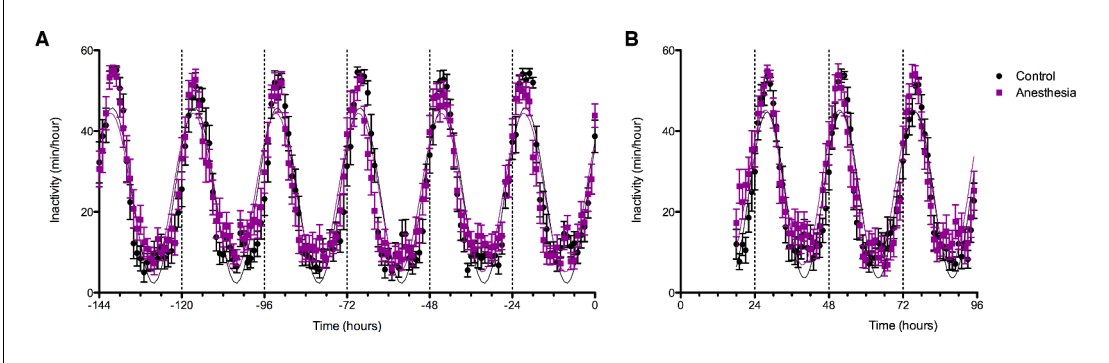

**Figure 5.** Effects of anesthetic exposure on rest-activity rhythms. (**A**) Rest activity plots are displayed in the week prior to the study day for volunteers that were subsequently randomized to anesthetized (purple) or control (black) conditions. (**B**) Rest-activity rhythms in the same participants are displayed on the evening of the study day and for the ensuing days. Time = 0 corresponds to midnight on the evening of the study day.

The online version of this article includes the following source data for figure 5:

**Source data 1.** Source data supporting *Figure 5*.

just before ROC (*Figure 4C*). Thus, both the local (PE) and global (LZC) metrics of cortical dynamics recover just as the brain is recovering consciousness. Similar state-dependent changes were observed despite different strategies in the parameter selection for PE (*Figure 4—figure supplement 1*), the threshold of binarization, and the method of generating surrogate data for LZC (*Figure 4—figure supplement 2*). Additionally, as expected, the non-anesthetized control group showed no differences among the seven resting-state eyes-closed epochs (*Figure 4—figure supplement 3*).

We explored further whether the EEG measures during EC1, Maintenance or pre-ROC were associated with the impairment of cognitive performance at emergence, where the maximal degree of impairment occurred. We found no evidence that the EEG measures during EC1 is associated with the impairment in accuracy or response time for all tasks (*Figure 4—figure supplement 4*). Of note, the LZC during Maintenance showed some correlation with the impairment of response time for MP (r = 0.522, p=0.004) and DSST (r = 0.446, p=0.016) tasks (*Figure 4—figure supplement 5*). Moreover, both the frontal PE and LZC were found to be weakly correlated with the impairment of accuracy for DSST task (r = 0.381, p=0.046 and r = 0.479, p=0.011) (*Figure 4—figure supplement 6*). These data must be interpreted with caution given the multiple statistical comparisons and weak correlations.

## Sleep-wake activity in the days following exposure to general anesthesia

Average rest-activity patterns for all study subjects are shown in *Figure 5A*. As expected, time of day significantly affected inactivity in both groups ($F_{129,\ 6867}$ = 46.24, p<0.0001). Cosinar analysis demonstrated that peak inactivity occurred between 3 and 4 am for both groups. Importantly, two-way ANOVA analysis revealed that over the week prior to the study day, there were no significant differences between the participants who would subsequently be anesthetized and those who would not ($F_{1,\ 6865}$ = 3.70, p>0.05). Moreover, there were no significant interactions between time and group ($F_{149,\ 6865}$ = 1.18, p>0.05). Analysis of rest activity resumed upon completion of the experiment on the study day. Actigraphy revealed a small yet statistically significant effect of anesthetic exposure, accounting for only 0.19% of the total variance ($F_{1,\ 3576}$ = 12.55, p=0.0004). This was attributable to an increase in inactivity in the isoflurane-exposed volunteers during the initial 2 hr of the early evening (*Figure 5B*). Activity patterns were not distinguishable after that time period. As before the study day, time remained as a highly significant predictor of inactivity accounting for 42% of the variance ($F_{77,\ 3576}$ = 35.78, p<0.0001).

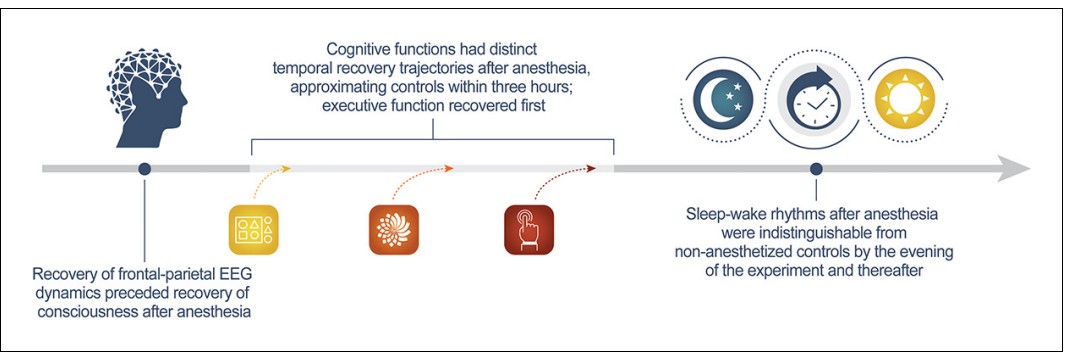

**Figure 6.** Summary of the study findings.

## Discussion

This is the most comprehensive and controlled study assessing cognitive recovery from the anesthetized state in healthy humans. Although there have been numerous studies of recovery from anesthetic-induced unconsciousness (*Långsjö et al., 2012*; *Purdon et al., 2013*; *Chennu et al., 2016*; *Kim et al., 2018*; *Banks et al., 2020*; *Scheinin et al., 2021*), our investigation was unique in that the length and depth of general anesthesia was consistent with surgical conditions but the experimental paradigm did not include any surgical intervention. This allowed us to study cognitive reconstitution after a major perturbation of arousal state while also informing the ongoing controversy regarding the effects of general anesthesia on human cognition. We have demonstrated that reconstitution of cognition after general anesthesia is a process that unfolds over time rather than a discrete, singular event. Cognitive recovery also follows a counterintuitive sequence. Executive function was found to recover early, whereas tests of processing speed, attention, and reaction time had a more prolonged recovery. Furthermore, EEG-based measures of cortical dynamics return to baseline just prior to the recovery of consciousness after general anesthesia, with entropy in frontal cortex statistically significantly higher than posterior cortex just after emergence. There were not, however, strong predictors of cognitive recovery based on EEG dynamics. Finally, sleep-wake activity patterns are essentially unperturbed after anesthetic exposure. These findings are summarized in *Figure 6*.

As described in the introduction, our hypothesis was motivated by preliminary evidence in the literature that neurocognitive recovery after general anesthesia proceeds in a caudal-to-rostral direction (*Långsjö et al., 2012*; *Reshef et al., 2019*), which would imply that executive functions mediated by anterior cortical structures would be the last to return. However, the finding that performance on the abstract matching test displayed more robust recovery than other cognitive functions indicates that there is early engagement of prefrontal cortex after recovery of consciousness. Furthermore, EEG analysis demonstrated that both anterior and posterior cortices were active in association with recovery of consciousness, although there was no statistically significant relationship between neurophysiologic complexity in prefrontal cortex and subsequent executive function performance. Collectively, our results are aligned with the theoretical framework of global neuronal workspace theory, which attempts to describe the neurocognitive mechanisms of conscious access (*Dehaene et al., 1998*; *Dehaene and Changeux, 2011*; *Mashour et al., 2020*). Access consciousness refers to the broad cognitive availability of sensory information, which often is measured based on report or task performance (as in this study), and is distinct from phenomenal consciousness, which has been defined as experience itself and hypothesized to be independent of post-perceptual processing in anterior cortex (*Block, 2005*; *Raccah et al., 2021*). According to workspace theory, a reverberant network that includes both prefrontal and posterior parietal cortex serves to sustain, amplify, and make available neural representations, with a 'broadcasting' effect that allows access by otherwise non-conscious cognitive processors (*Mashour et al., 2020*). The active frontal-parietal networks at return of consciousness and early recovery of a cognitive test associated with dorsolateral prefrontal cortex are consistent with at least a partially reintegrated global neuronal workspace. However, a fully intact workspace requires exquisite temporal coordination. We hypothesize that the differential temporal recovery of cognitive functions is a manifestation of differential temporal dynamics in the neural structures and cognitive processors across the workspace, which appear to

return to, or be asymptotically close to, baseline approximately 3 hr after recovery of consciousness. It is important to note that the first evidence of conscious access during recovery was hand squeezing to command, which is a simple, binary task unlike the more challenging cognitive battery with continuous performance measures such as accuracy and speed. Thus, hand squeezing to command would not be robust evidence for a full reintegration of the global neuronal workspace. Further work is needed to determine if global neuronal workspace theory is sufficient to describe the reconstitution of consciousness and cognition after a major perturbation such as a general anesthesia, or whether other cognitive theories need to be employed.

This study is consistent with our prior analysis of source-localized alpha oscillations and graph-theoretical variables such as global efficiency and modularity, which were identified in a subset of participants who had high-density EEG and which also recovered within 3 hr after emergence from general anesthesia (*Blain-Moraes et al., 2017*). From a clinical perspective, it is remarkable that within 3 hr of recovery after a prolonged and deep general anesthetic, participants were performing a variety of complex cognitive tasks with similar accuracy and speed in comparison to participants who had not been anesthetized. On a shorter timescale, both local and global dynamic measures of cortical activity returned to baseline levels just prior to the return of responsiveness after general anesthesia. On a longer timescale, there was no evidence of disrupted sleep in the days following anesthetic exposure compared to non-anesthetized controls. This latter finding is striking considering that actigraphy measures of sleep are sensitive to stress, low alcohol exposure, and even sedative effects of non-alcoholic beer (*Mezick et al., 2009*; *Geoghegan et al., 2012*; *Franco et al., 2012*). The data suggest that, in healthy humans, higher cognition and arousal states recover uneventfully after deep general anesthesia.

The recovery of cortical dynamics in anterior and posterior cortical areas just prior to the return of consciousness in our experimental paradigm should be considered in light of prior studies focused on emergence from general anesthesia. One study of healthy volunteers anesthetized with either propofol or dexmedetomidine and imaged using positron emission tomography found that those responsive to command exhibited activation of subcortical arousal centers with only limited frontal-parietal cortex involvement (*Långsjö et al., 2012*) a recent study with a more sophisticated pharmacologic dosing strategy has largely confirmed these findings (*Scheinin et al., 2021*). These data are consistent with a study of functional magnetic resonance imaging that demonstrated a transient activation of brainstem loci upon emergence from propofol sedation (*Nir et al., 2019*) as well as a positron emission tomography study revealing subcortical and posterior cortical activation in association with the reversal of propofol sedation using the acetylcholinesterase inhibitor physostigmine (*Xie et al., 2011*). However, it is important to note that all of these studies involved experimental conditions in which there was ongoing exposure to a sedative-hypnotic, coupled with pharmacological or behavioral stimulation. By contrast, our paradigm reflects spontaneous emergence with residual isoflurane levels predicted to be 1 to 4 orders of magnitude below those required for hypnosis, which likely accounts for evidence of the robust return of cortical dynamics. However, it is worth noting that—in addition to the subcortical sites identified in human and animal studies (*Kelz et al., 2019*) —the prefrontal cortex might play a critical role in the control of arousal states of relevance to general anesthesia. One animal study demonstrated that cholinergic stimulation of medial prefrontal cortex, but not two sites in posterior parietal cortex, was sufficient to reverse the anesthetized state despite continuous administration of clinically relevant concentrations of sevoflurane (*Pal et al., 2018*). Thus, the prefrontal cortex might play a critical role in recovery of both consciousness (medial prefrontal) and cognition (dorsolateral prefrontal).

Strengths of this study include: (1) surgically relevant anesthetic concentrations and duration of anesthetic exposure, in the absence of the confound of surgery itself; (2) multicenter design with substantially more participants than are typically found in studies of anesthetic-induced unconsciousness; (3) a non-anesthetized population to control for learning effects of cognitive assessment as well as comparison of sleep-wake patterns; and (4) complement of neurophysiological, cognitive, and behavioral measures on different time scales. Limitations of this study that constrain interpretation include: (1) young healthy population that precludes extrapolation of findings to older, younger, or sicker surgical patients encountered in clinical care; (2) only one anesthetic regimen tested, albeit a clinically common one, with unclear relevance to other perturbations such as sleep or pathological disorders of consciousness; (3) inability to blind participants to anesthetized and non-anesthetized conditions, which could potentially influence results; (4) relatively sparse EEG channels that were

found to be a reliable sources of data across all participants; (5) no assessment of sleep macroarchitecture (i.e. rapid eye movement sleep vs. slow-wave sleep); (6) no longer term cognitive or behavioral assessment beyond the first three days of post-anesthetic actigraphy; and (7) inability to assess source-localized brain regions, subcortical regions, or resting-state networks.

In conclusion, this study establishes neurophysiologic, cognitive, and behavioral recovery patterns after a surgically relevant general anesthetic in human volunteers. The rapid recovery of cortical dynamics just prior to recovering consciousness, the restored accuracy of executive function and multiple cognitive functions within 3 hr of emergence, and the normal sleep-wake patterns in the days following the experiment provide compelling evidence that the healthy brain is resilient to the effects of even deep general anesthesia. The findings also suggest that the immediate and persistent cognitive dysfunction identified after general anesthesia in healthy animals (*Culley et al., 2004*; *Valentim et al., 2008*; *Carr et al., 2011*; *Callaway et al., 2012*; *Zurek et al., 2012*; *Jevtovic-Todorovic et al., 2013*; *Zurek et al., 2014*; *Jiang et al., 2017*) does not necessarily translate to healthy humans and that postoperative neurocognitive disorders might relate to factors other than general anesthesia, such as surgery or patient comorbidity (*Wildes et al., 2019*; *Krause et al., 2019*). Finally, our data are consistent with the global neuronal workspace theory of conscious access.

## Acknowledgements

This study was funded by a collaborative grant from the James S McDonnell Foundation, St. Louis, MO; National Institutes of Health (Bethesda, MD, USA) grant T32GM112596; and the anesthesiology departments of the University of Michigan, University of Pennsylvania, and Washington University.

## Additional information

### Funding

| Funder | Grant reference number | Author |
|---|---|---|
| James S. McDonnell Foundation | Understanding Human Cognition | George A Mashour<br>Michael S Avidan<br>Max B Kelz |
| National Institutes of Health | T32GM112596 | Andrew R McKinstry-Wu |
| University of Michigan | | George A Mashour |
| University of Pennsylvania | | Max B Kelz |
| Washington University in St. Louis | | Michael S Avidan |

The funders had no role in study design, data collection and interpretation, or the decision to submit the work for publication.

### Author contributions

George A Mashour, Conceptualization, Data curation, Funding acquisition, Investigation, Writing - original draft, Project administration; Ben JA Palanca, Stefanie Blain-Moraes, Data curation, Investigation, Writing - review and editing; Mathias Basner, Conceptualization, Formal analysis; Duan Li, Formal analysis, Methodology, Writing - review and editing; Wei Wang, Formal analysis; Nan Lin, Formal analysis, Investigation, Writing - review and editing; Kaitlyn Maier, Maxwell Muench, Hannah Maybrier, Project administration; Vijay Tarnal, Giancarlo Vanini, E Andrew Ochroch, Rosemary Hogg, Marlon Schwartz, Randall Hardie, Ellen Janke, Goodarz Golmirzaie, Data curation; Paul Picton, Andrew R McKinstry-Wu, Data curation, Writing - review and editing; Michael S Avidan, Conceptualization, Data curation, Funding acquisition, Investigation, Writing - original draft; Max B Kelz, Conceptualization, Data curation, Formal analysis, Funding acquisition, Writing - original draft

## Author ORCIDs
George A Mashour (ID) https://orcid.org/0000-0001-5457-5932
Andrew R McKinstry-Wu (ID) http://orcid.org/0000-0001-7078-4603
Max B Kelz (ID) http://orcid.org/0000-0002-2803-6078

## Ethics
Clinical trial registration NCT01911195.
Human subjects: The study received ethics committee approval from the University of Michigan, Washington University, and the University of Pennsylvania; written informed consent was obtained after careful discussion with each participant.

## Decision letter and Author response
Decision letter https://doi.org/10.7554/eLife.59525.sa1
Author response https://doi.org/10.7554/eLife.59525.sa2

# Additional files

## Supplementary files
• Supplementary file 1. A reports the sample size of data epochs for electroencephalographic analysis and file 1B describes model selection for the statistical analysis of electroencephalographic measures.

• Transparent reporting form

## Data availability
All data generated or analyzed during this study are included in the manuscript and supporting files. Source data have been provided for Figures 2-5.

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
