## [Decision Letter]

**Acceptance summary:**

This paper is concisely written with a conceptual structure that is easy to follow.

The study is well controlled, and uses a wide range of neurocognitive tests to assess different aspects of cognition. The main findings, that executive function recovers before other potentially more basic aspects of cognition, supported by a similarly early return of frontal cortical dynamics offer important insights into the recovery of cognition after pharmacologically induced unconsciousness. These findings are novel and, although cannot be generalised beyond anaesthetic agent isoflurane, will be of interest to clinical anaesthesiologists, healthy individuals undergoing isoflurane-based general anaesthesia, and researchers investigating the relationship between consciousness and cognition.

**Decision letter after peer review:**

Thank you for submitting your article "Recovery of consciousness and cognition after general anesthesia in humans" for consideration by *eLife*. Your article has been reviewed by 3 peer reviewers, including Redmond G O’Connell as the Reviewing Editor and Reviewer #1, and the evaluation has been overseen by Richard Ivry as the Senior Editor. The following individual involved in review of your submission has agreed to reveal their identity: Bekinschtein Tristan (Reviewer #2).

The reviewers have discussed the reviews with one another and the Reviewing Editor has drafted this decision to help you prepare a revised submission.

As the editors have judged that your manuscript is of interest, but as described below that significant revisions are required before it is published, we would like to draw your attention to changes in our revision policy that we have made in response to COVID-19 (https://elifesciences.org/articles/57162). First, because many researchers have temporarily lost access to the labs, we will give authors as much time as they need to submit revised manuscripts. We are also offering, if you choose, to post the manuscript to bioRxiv (if it is not already there) along with this decision letter and a formal designation that the manuscript is "in revision at *eLife*". Please let us know if you would like to pursue this option. (If your work is more suitable for medRxiv, you will need to post the preprint yourself, as the mechanisms for us to do so are still in development.)

Summary

The three reviewers agreed that the paper reports results that are important as they appear to offer novel insights into the dynamics of cognitive recovery following loss of consciousness; an area that has been relatively under-investigated to date. However all three reviewers also highlighted some significant concerns regarding aspects of the study rationale and methodology. A consolidated list of major points is provided below:

Essential Revisions:

1. The authors do not provide any rationale for their starting hypotheses regarding the sequence in which cognitive functions are expected to recover. For example, the prediction that vigilant attention should recover first is not intuitive and no basis or prior literature is provided for this prediction.

2. In the Introduction the authors use terms that are difficult to identify with established cognitive constructs e.g. top of page 5 it is stated that 'attention' should recover first but this is an extremely loose term that incorporates a range of different subcomponents (vigilant/selective/spatial etc). Similarly, 'scanning and tracking' does not refer to a cognitively- or psychologically-motivated distinct function (top of page 5). The authors should also clarify which tests are being used to measure which functions because it is currently confusing that 5 functions are being linked to 6 behavioural tests. The descriptions in the methods section do not help to clarify the relationships; e.g., the Motor Praxis Task (MP) task linked to complex scanning and visual tracking in the introduction, is described to measure sensorimotor speed.

3. In general, the three reviewers agreed that many aspects of the methodology are too vaguely described and require clarification so that the study can be understood by the reader and potentially replicated by other labs in the future. Some critical examples are highlighted in the comments below but more generally, if the authors prefer to position the Results before the Methods then they should ensure that there is sufficient detail in the Results to allow the reader to understand the experiment. For example, the reader should not have to wait to reach the Methods to be informed that there were two separate groups and that the control group exercised prior to cognitive testing. For example, the results in 2nd paragraph of page 7 are very scantly described, and a summary table with full disclosure of test statistic values is needed. Figures 2 and 3 lack signposting of statistical significance, a missed opportunity given the rich information provided. E.g., it's impossible to visualise when performance in each task recovers. Similarly, the sentence "The PE demonstrated significant differences associated with behavioural states (F9, 86 = 42.423, p<0.001) ,brain regions (F1, 257 = 4.275, p=0.040) and the interaction between them (F9, 85 = 2.750, p=0.007)." does not tell the reader what test was run or what factors were included in the statistical model and so it is impossible for the reader to fully appreciate the meaning of this result. In general, there is a quite a lot of work to be done throughout the manuscript to render the description of analyses and results sufficiently transparent, and comprehensive.

4. The critical term 'recovery', on which the key results hinge, is never clearly defined or operationalised. A Bayesian regression approach is vaguely described in the Methods section but the information provided does not explain how recovery is actually defined or established. Currently, the study appears to rely heavily on null-hypothesis significance testing to draw its conclusions but p>0.05 does not mean that we should embrace the null hypothesis. The authors should consider including tests (e.g. Bayes Factor) to determine the degree of statistical confidence that post-sedation test scores no longer differ reliably from those at baseline or else provide a strong justification for their current approach.

5. As the authors themselves note, the potential for practice effects to confound any recovery estimates is a critical concern here one that needs to be addressed in a revision. Relatedly, there is the concern that these cognitive tests may differ quite markedly in their difficulty for potentially trivial reasons and this could impact on estimates of recovery. For example, if performance on some tasks is close to ceiling then recovery may be deemed to occur earlier. Can the authors provide new analyses that would address the possibility that some of these tasks may simply be more sensitive to cognitive perturbations than others?

6. The EEG analyses are potentially interesting but the authors do not provide any rationale for focussing in on these particular metrics out of the many that they could have considered (e.g. connectivity, power). Additional text discussing these metrics should be provided along with hypotheses regarding their post-sedation dynamics. In addition, the fact that the EEG trends are never linked to the cognitive ones limits the conclusions that can be drawn here. Did the authors consider examining the degree to which these EEG changes are predictive of cognitive recovery across subjects? Did they consider examining how levels of complexity before and/or during sedation might influence the degree or speed of recovery?

[Editors' note: further revisions were suggested prior to acceptance, as described below.]

Thank you for resubmitting your work entitled "Recovery of consciousness and cognition after general anesthesia in humans" for further consideration by *eLife*. Your revised article has been reviewed by two peer reviewers, one of whom is a member of our Board of Reviewing Editors, and the evaluation has been overseen by Richard Ivry as the Senior Editor.

The manuscript has been substantially improved but there are some remaining issues that need to be addressed, as outlined below. In particular, there is a need to flesh out the study rationale, to clarify why these particular EEG metrics were considered, to further clarify the theoretical background to the study and to interpret the results with respect to extant cognitive theories. The two reviewers have provided guidance on each of these points. I think these further changes, which would really help to ensure that the paper reaches and resonates with the widest possible audience.

The specific comments made by the two reviewers are provided below.

*Reviewer #1:*

I find the manuscript much improved. In particular I think placing the Methods ahead of the Results works well in this case and the authors have provided improved clarity regarding the study goals. I do have some comments regarding the written presentation:

The Abstract opens on a very vague note. I don't think it will be apparent to many readers what the authors mean when they say that understanding the re-emergence of cognition is important 'neurobiologically'. This is clarified in the Introduction but I think the Abstract needs to provide a more clear statement regarding the study rationale in order to ensure that the article will be read by a wide readership. For example, the question of whether recovery is progressive and whether certain cognitive functions recover before others could be stated in the second sentence. Extremely vague opening line. Add the hypotheses here.

I also take issue with the following statement in the Abstract: 'Contrary to our hypothesis, executive function returned first, with frontal cortical dynamics recovering faster than posterior cortical dynamics.' This phrasing implies that a correlation between the EF recovery and posterior cortical dynamics has been established in the author's analyses which in fact is not the case. The authors should rephrase this sentence to avoid giving that impression e.g. deal with the cognitive and EEG results in separate sentences.

I think the authors need to provide the reader with more information regarding the types of EEG metrics that the authors have chosen to analyse. Why these particular metrics and not others? Despite being an EEG researcher myself I found the term 'local and global dynamics' very vague and unclear. Is the idea to establish whether a particular brain area has recovered normal levels of activity? Some clear statement regarding the functional importance of the chosen metrics is necessary.

*Reviewer #2:*

Thanks for amending the manuscript, I think that there is still some work to do in the depiction of results front, make them easy to read and interpretable; on the theoretical framework, making a bit more explicit during the interpretation what theoretical tools are you using to discuss the results; to interpret the correlations with caution.

I think that the information of the cognitive domains that are tested with the tasks is key and forms an important aspect of the paper and the authors should clearly frame the initial hypotheses and results in theoretical terms for the cognitive neuroscience colleagues who will be reading the paper. It will also help understanding better the aims of this work outside the clinical description. From a cognitive perspective, what is recovery from unconsciousness? Can we frame different cognitive recovery dynamics in terms of the information integration that networks regain? How is that measured or conceptualized? Is this theory-driven account captured in any shape or form by the complexity measures?

It has been shown that primary cortices (auditory, visual, motor) recover faster or are more resilient to consciousness challenges (sleep, sedation). Does this inform the expected recovery dynamics or helps interpret it?

Can the tasks be ordered in terms of the cognitive demands they pose to the system (the brain)? Does this help to frame the recovery results found?

Do the authors think of these results in light of cognitive demands, information processing, integration of features, perceptual and cognitive modules? Please think in the interpretation of your results in light of a theoretical framework and whether it contributes or not to theory itself. Is this experimental model of cognitive recovery complementary to classic cognitive interference, non-conscious drug challenges, Transcranial magnetic or electrical stimulation?

In short, I do not think it is enough to restrict the introduction or discussion to the cognitive test and its putative neural correlates for the reasons and guiding questions above.

The other comment I have is the depiction of the results. A graph overlapping (even if simplified) the PP speed distributions and the accuracy distributions would make a great figure that to be used for other authors to reference this work in talks and to be used in teaching. In its current form it requires a lot of explanation and it does not paint in one look the findings. This is a preference and nice to have so the work can be featured better in the future.

For the correlations results, there are clearly weak or no associations and given the number of participants (low for correlations) and high number of comparison the likelihood of chance association is dangerously high. A word of caution in the paper about this will help and also please check if the even the correction for multiple correlations also renders this apparent significant associations void.

Will the behavioural and EEG data become available soon and if so in which platform.

---

## [Author Response]

Essential Revisions:1. The authors do not provide any rationale for their starting hypotheses regarding the sequence in which cognitive functions are expected to recover. For example, the prediction that vigilant attention should recover first is not intuitive and no basis or prior literature is provided for this prediction.

Our primary hypothesis, described a priori in a published protocol, is that emergence occurs through a *process*, not at a *point*. Importantly, this hypothesis is independent of predicting the specific sequence of recovery. We have now stated this primary hypothesis more clearly in the introduction. In the revision, we have been more conservative in formulating the sequence of recovery, stating that – pursuant to our hypothesis regarding process – arousal and recovery of consciousness would occur first and return of higher executive function would return last. This latter point is defensible because there is neurologic examination and neuroimaging evidence to suggest that recovery from anesthesia occurs in a caudal to rostral manner (e.g., Langsjo et al., J Neuroscience, 2012;32(14):4935-43, and Reshef et al., Anesthesiology, 2019;130(3):462-471). In the revised version, we state that executive function – measured using a test that is known to engage the rostral structure dorsolateral prefrontal cortex – was predicted to recover last. This is perfectly consistent with our original formulation and statistical testing but avoids less defensible granularity in the recovery sequence of cognitive functions.

2. In the Introduction the authors use terms that are difficult to identify with established cognitive constructs e.g. top of page 5 it is stated that 'attention' should recover first but this is an extremely loose term that incorporates a range of different subcomponents (vigilant/selective/spatial etc). Similarly, 'scanning and tracking' does not refer to a cognitively- or psychologically-motivated distinct function (top of page 5). The authors should also clarify which tests are being used to measure which functions because it is currently confusing that 5 functions are being linked to 6 behavioural tests. The descriptions in the methods section do not help to clarify the relationships; e.g., the Motor Praxis Task (MP) task linked to complex scanning and visual tracking in the introduction, is described to measure sensorimotor speed.

We have restricted our description in the introduction to the cognitive tests rather than the cognitive domains, in order to avoid imprecision in the description of the cognitive function or mapping to the actual cognitive test. This is defensible and grounded in the use of this particular neurocognitive battery for assessment of cognitive deficits related to perturbations of other arousal states (i.e., recovery from sleep deprivation). We have also included a table (Author response table 1) that gives an overview of the tests, associated cognitive domain, and associated brain region thought to support the domain.

**Author response table 1. resptable1:** 

Test	Cognitive Domains Assessed	Brain Regions Primarily Recruited
Motor Praxis	Sensory-motor speed	Sensorimotor cortex
Visual Object Learning	Spatial learning and memory	Medial temporal cortex, hippocampus
Fractal 2-Back	Working memory	Dorsolateral prefrontal cortex, cingulate, hippocampus
Abstract Matching	Abstraction, concept formation	Prefrontal cortex
Digit Symbol Substitution	Complex scanning and visual tracking, working memory	Temporal cortex, prefrontal cortex, motor cortex
Psychomotor Vigilance	Vigilant attention	Prefrontal cortex, motor cortex, inferior parietal and some visual cortex

3. In general, the three reviewers agreed that many aspects of the methodology are too vaguely described and require clarification so that the study can be understood by the reader and potentially replicated by other labs in the future. Some critical examples are highlighted in the comments below but more generally, if the authors prefer to position the Results before the Methods then they should ensure that there is sufficient detail in the Results to allow the reader to understand the experiment. For example, the reader should not have to wait to reach the Methods to be informed that there were two separate groups and that the control group exercised prior to cognitive testing. For example, the results in 2nd paragraph of page 7 are very scantly described, and a summary table with full disclosure of test statistic values is needed. Figures 2 and 3 lack signposting of statistical significance, a missed opportunity given the rich information provided. E.g., it's impossible to visualise when performance in each task recovers. Similarly, the sentence "The PE demonstrated significant differences associated with behavioural states (F9, 86 = 42.423, p<0.001) ,brain regions (F1, 257 = 4.275, p=0.040) and the interaction between them (F9, 85 = 2.750, p=0.007)." does not tell the reader what test was run or what factors were included in the statistical model and so it is impossible for the reader to fully appreciate the meaning of this result. In general, there is a quite a lot of work to be done throughout the manuscript to render the description of analyses and results sufficiently transparent, and comprehensive.

We have, in response to this helpful suggestion, repositioned the Methods section before the Results section. We have also reviewed the manuscript critically for areas in which methodology and the ensuing explanation of our results could be made more explicit. The first line of the Materials and methods section highlights our published protocol, which contains more details and is open access.

*4. The critical term 'recovery', on which the key results hinge, is never clearly defined or operationalised. A Bayesian regression approach is vaguely described in the Methods section but the information provided does not explain how recovery is actually defined or established. Currently, the study appears to rely heavily on null-hypothesis significance testing to draw its conclusions but p>0.05 does not mean that we should embrace the null hypothesis. The authors should consider including tests (e.g. Bayes Factor) to determine the degree of statistical confidence that post-sedation test scores no longer differ reliably from those at baseline or else provide a strong justification for their current approach.*

We thank the reviewers for this critically important point. The recovery time involves both a performance speed and performance accuracy. Both processes are defined by the time for anesthetized subjects to have the test score back to their respective speed/accuracy baseline scores. We have revised the methods and results to include the following information/analysis.

Since our model was yt=ybaseline+α+β∗et,the recovery time was calculated by T=log⁡(−αβ)/.Author response image 1 shows the histogram of recovery time (measured in hours) for *speed* in each testing domain based on 10000 Markov Chain Monte Carlo samples.

As indicated both by the Y-axis values and by the simulated distribution, the recovery of NBCK (as measured by speed to task completion) is strongly impaired. Most samples did not recover over the experimental time course – a fact mirrored in the simulations.For the other 5 tests, the posterior probability that:

a. Recovery of MP speed occurred more than 0.5 hour before AM, PVT and DSST are 54%, 91% and 83%, respectively; the 90% corresponding credible intervals of the difference in recovery time are (-1.31, 0.24), (-3.95, -0.02) and (-1.64, -0.24);

b. Recovery of VOLT speed occurred more than 0.5 hour before MP, AM, PVT and DSST are 82%, 98%, 100% and 100%, respectively; the 90% corresponding credible intervals of the difference in recovery time are (-1.97, -0.12), (-2.35, -0.79), (-4.84, -1.17) and (-2.66, -1.33);

c. Recovery of AM speed occurred more than 0.5 hour before PVT and DSST are 79% and 40%, respectively; the 90% corresponding credible intervals of the difference in recovery time are (-3.49, 0.10) and (-0.92, 0.10);

d. Recovery of DSST occurred more than 0.5 hour before PVT is 64%; the 90% corresponding credible interval of the difference in recovery time is (-3.02, 0.47).

Author response image 2 shows the histogram of recovery time in hour for *accuracy* in each testing domain based on 10000 Markov Chain Monte Carlo samples.

**Author response image 2. respfig2:** 

The accuracy in PVT is strongly impaired as most samples cannot recover.For the other 5 tests, the posterior probability that

a. Recovery of MP occurred more than 0.5 hour before VOLT, NBCK and DSST are 93%, 97% and 99%, respectively; the 90% corresponding credible intervals of the difference in recovery time are (-2.82, -0.27), (-3.6, -0.77) and (-0.15, 0.78);

b. Recovery of VOLT occurred more than 0.5 hour before NBCK and DSST are 55% and 50%, respectively; the 90% corresponding credible intervals of the difference in recovery time are (-2.38, 1.32) and (-1.69, 0.74);

c. Recovery of AM occurred more than 0.5 hour before VOLT, NBCK and DSST are 98%, 99% and 100%, respectively; the 90% corresponding credible intervals of the difference in recovery time are (-2.92, -0.47), (-3.66, -0.91) and (-2.47, -1.15);

d. Recovery of DSST occurred more than 0.5 hour before NBCK is 42%; the 90% corresponding credible interval of the difference in recovery time is (-1.55, 1.09).

5. As the authors themselves note, the potential for practice effects to confound any recovery estimates is a critical concern here one that needs to be addressed in a revision. Relatedly, there is the concern that these cognitive tests may differ quite markedly in their difficulty for potentially trivial reasons and this could impact on estimates of recovery. For example, if performance on some tasks is close to ceiling then recovery may be deemed to occur earlier. Can the authors provide new analyses that would address the possibility that some of these tasks may simply be more sensitive to cognitive perturbations than others?

We agree that it is critical for the interpretation of the results to carefully take practice effects into account. All six cognitive tests with the exception of PVT are known to have protracted practice effects (Basner et al., J Clin Exp Neuropsychol, 2020;42(5): 516-29). Therefore, having an equally sized non-anesthetized control group is a major strength of this study. Without this control group, we would not have been able to establish a “return to normal,” as the “normal” would be drifting to faster/more accurate performance with repeated administration.

In Author response table 2, when corrected for multiple comparisons, our adjusted p-value cutoff considered to be significant is p < 0.05/12 = 0.0042. Hence, no baseline significant differences exist between the experimental and the control group. Furthermore, after only two administrations, neither group hit a ceiling (i.e., accuracy is below 100%) or a floor. When testing the PVT and DSST, accuracy is always close to 100% in alert and motivated subjects. Consequently, the focus of these tests is speed and cognitive throughput. This does not mean the tests are easy and they certainly did not recover first. The reviewers are correct that it is possible that some of the tasks may be more sensitive to cognitive perturbations than others. However, this relationship is not straightforward. For example, even when considering a single test – the MP test shows the most profound drop in performance at the point of emergence, falling by more than 10 z-scores from baseline pre-anesthesia performance. Nevertheless, accuracy on this test drops minimally immediately upon emergence from anesthesia, but fully recovers within 30 minutes unlike MP speed.

**Author response table 2. resptable2:** 

	Control	Experimental			
Variable	Average BL	SD BL	Average BL	SD BL	p-value
MP Average RT [ms]	962.4	155.1	1049.5	172.3	0.0440
MP Accuracy	28.8%	9.3	30.0%	10.3%	0.6527
VOLT Average RT [ms]	2067.0	390.6	2088.8	381.7	0.8271
VOLT Correct	80.3%	13.5%	83.7%	10.6%	0.2919
NBCK Average RT [ms]	605.8	92.4	589.3	89.1	0.4847
NBCK Correct	85.9%	7.4%	85.7%	6.3%	0.9040
AM Average RT [ms]	2116.4	661.7	1895.0	504.3	0.1504
AM Correct	68.3%	13.6%	68.8%	11.5%	0.8918
PVT Speed [1/s]	5.6	0.3	5.7	0.3	0.3889
PVT Accuracy	93.6%	4.2%	93.1%	4.6%	0.6922
DSST Average RT [ms]	1417.5	272.9	1419.6	234.6	0.9752
DSST Correct	95.7%	7.6%	97.2%	3.2%	0.3331

6. The EEG analyses are potentially interesting but the authors do not provide any rationale for focussing in on these particular metrics out of the many that they could have considered (e.g. connectivity, power). Additional text discussing these metrics should be provided along with hypotheses regarding their post-sedation dynamics. In addition, the fact that the EEG trends are never linked to the cognitive ones limits the conclusions that can be drawn here. Did the authors consider examining the degree to which these EEG changes are predictive of cognitive recovery across subjects? Did they consider examining how levels of complexity before and/or during sedation might influence the degree or speed of recovery?

We have provided justification in the revised manuscript for the choice of permutation entropy and Lempel-Ziv complexity. Furthermore, to explore whether the EEG measures are predictive of cognitive recovery in the post-anesthetic period, we assessed the associations of EEG measures during the pre-anesthetic baseline (EC1), the intra-anesthetic period (Maintenance) and the period immediately preceding recovery of responsiveness (pre-ROC) with the impairment of cognitive performance at emergence (just after recovery of consciousness). Spearman’s rank correlation analysis showed no association in EEG measures during EC1, but weak correlations were found between the LZC during Maintenance and the impairment of response time for MP (Motor Praxis) (r=0.522, p=0.004) and DSST (Digit Symbol Substitution Test) tasks (r=0.446, p=0.016), as well as between frontal PE and LZC during pre-ROC and the impairment of accuracy for DSST task (r=0.381, p=0.046 and r=0.479, p=0.011). We have provided these results in Figure 4—figure supplements 4-6.

[Editors' note: further revisions were suggested prior to acceptance, as described below.]

Reviewer #1:I find the manuscript much improved. In particular I think placing the Methods ahead of the Results works well in this case and the authors have provided improved clarity regarding the study goals. I do have some comments regarding the written presentation:The Abstract opens on a very vague note. I don't think it will be apparent to many readers what the authors mean when they say that understanding the re-emergence of cognition is important 'neurobiologically'. This is clarified in the Introduction but I think the Abstract needs to provide a more clear statement regarding the study rationale in order to ensure that the article will be read by a wide readership. For example, the question of whether recovery is progressive and whether certain cognitive functions recover before others could be stated in the second sentence. Extremely vague opening line. Add the hypotheses here.

Thank you for highlighting these opportunities for improvement. We have revised the abstract, also addressing comments regarding theoretical framework from Reviewer 2:

“Understanding how the brain recovers from unconsciousness can inform neurobiological theories of consciousness and guide clinical investigation. […] Early engagement of prefrontal cortex in recovery of consciousness and cognition is consistent with global neuronal workspace theory.”

I also take issue with the following statement in the Abstract: 'Contrary to our hypothesis, executive function returned first, with frontal cortical dynamics recovering faster than posterior cortical dynamics.' This phrasing implies that a correlation between the EF recovery and posterior cortical dynamics has been established in the author's analyses which in fact is not the case. The authors should rephrase this sentence to avoid giving that impression e.g. deal with the cognitive and EEG results in separate sentences.

We agree with the reviewer’s interpretation and (as can be seen in the abstract copied above) have separated the concepts in two sentences.

I think the authors need to provide the reader with more information regarding the types of EEG metrics that the authors have chosen to analyse. Why these particular metrics and not others? Despite being an EEG researcher myself I found the term 'local and global dynamics' very vague and unclear. Is the idea to establish whether a particular brain area has recovered normal levels of activity? Some clear statement regarding the functional importance of the chosen metrics is necessary.

In the revised manuscript we have avoided the phrase “local and global dynamics” in the introduction and instead state:

“…all participants had electroencephalographic recording throughout the experiment to assess cortical dynamics of relevance to consciousness and cognition, with techniques that have been used to assess information processing in specific brain regions (permutation entropy) as well as more complex spatiotemporal patterns across the cortex (Lempel-Ziv complexity).”

Furthermore, in the last revision, we added rationale for the selected analytic techniques and highlighted why we chose not to employ phase-based connectivity measures or graph-theoretical variables.

Reviewer #2:Thanks for amending the manuscript, I think that there is still some work to do in the depiction of results front, make them easy to read and interpretable; on the theoretical framework, making a bit more explicit during the interpretation what theoretical tools are you using to discuss the results; to interpret the correlations with caution.I think that the information of the cognitive domains that are tested with the tasks is key and forms an important aspect of the paper and the authors should clearly frame the initial hypotheses and results in theoretical terms for the cognitive neuroscience colleagues who will be reading the paper. It will also help understanding better the aims of this work outside the clinical description. From a cognitive perspective, what is recovery from unconsciousness? Can we frame different cognitive recovery dynamics in terms of the information integration that networks regain? How is that measured or conceptualized? Is this theory-driven account captured in any shape or form by the complexity measures?It has been shown that primary cortices (auditory, visual, motor) recover faster or are more resilient to consciousness challenges (sleep, sedation). Does this inform the expected recovery dynamics or helps interpret it?Can the tasks be ordered in terms of the cognitive demands they pose to the system (the brain)? Does this help to frame the recovery results found?Do the authors think of these results in light of cognitive demands, information processing, integration of features, perceptual and cognitive modules? Please think in the interpretation of your results in light of a theoretical framework and whether it contributes or not to theory itself. Is this experimental model of cognitive recovery complementary to classic cognitive interference, non-conscious drug challenges, Transcranial magnetic or electrical stimulation?In short, I do not think it is enough to restrict the introduction or discussion to the cognitive test and its putative neural correlates for the reasons and guiding questions above.

This was an important recommendation and we appreciate the opportunity to broaden the theoretical framework. In response, we have added more than a page to the Discussion to frame the interpretation of our results using the Global Neuronal Workspace (GNW) theory. We acknowledge that there are a number of options for theoretical frameworks but we have chosen this framework because our study focused on both consciousness and cognition, and also suggested a role for prefrontal cortex in recovery of consciousness and cognition, so it aligns with GNW as a cognitive theory of conscious access. Because our initial hypothesis was motivated by empirical rather than theoretical considerations, we restricted the application of GNW theory to a discussion of the results. The following appears after the first paragraph in the Discussion:

“As described in the introduction, our hypothesis was motivated by preliminary evidence in the literature that neurocognitive recovery after general anesthesia proceeds in a caudal-to-rostral direction Långsjö et al., 2012; Reshef et al., 2019(; ), which would imply that executive functions mediated by anterior cortical structures would be the last to return. […] Further work is needed to determine if global neuronal workspace theory is sufficient to describe the reconstitution of consciousness and cognition after a major perturbation such as a general anesthesia, or whether other cognitive theories need to be employed.”

The other comment I have is the depiction of the results. A graph overlapping (even if simplified) the PP speed distributions and the accuracy distributions would make a great figure that to be used for other authors to reference this work in talks and to be used in teaching. In its current form it requires a lot of explanation and it does not paint in one look the findings. This is a preference and nice to have so the work can be featured better in the future.

Thank you for this excellent suggestion. In response, we have created an additional figure (Figure 6; referenced in the Discussion) that summarizes the main findings regarding the recovery of consciousness, recovery of cognition, and recovery of sleep-wake patterns.

For the correlations results, there are clearly weak or no associations and given the number of participants (low for correlations) and high number of comparison the likelihood of chance association is dangerously high. A word of caution in the paper about this will help and also please check if the even the correction for multiple correlations also renders this apparent significant associations void.

We agree. We have added a note of caution regarding these findings:

“These data must be interpreted with caution given the multiple statistical comparisons and weak correlations.”

Will the behavioural and EEG data become available soon and if so in which platform.

In version 2/revision 1 of the manuscript, we included formatted data sets that support each figure.